# The Effect Ultrasound and Surfactants on Nanobubbles Efficacy against *Listeria innocua* and *Escherichia coli* O157:H7, in Cell Suspension and on Fresh Produce Surfaces

**DOI:** 10.3390/foods10092154

**Published:** 2021-09-12

**Authors:** Shamil Rafeeq, Reza Ovissipour

**Affiliations:** 1FutureFoods Lab, Virginia Seafood Agricultural Research and Extension Center, Virginia Tech, Hampton, VA 23669, USA; shamilrafeeq363@gmail.com; 2Department of Food Science and Technology, Virginia Tech, Blacksburg, VA 24060, USA

**Keywords:** nanobubbles, ultrasound, surfactant, cell suspension, spinach leaves

## Abstract

Removing foodborne pathogens from food surfaces and inactivating them in wash water are critical steps for reducing the number of foodborne illnesses. In this study we evaluated the impact of surfactants on enhancing nanobubbles’ efficacy on *Escherichia coli* O157:H7, and *Listeria innocua* removal from spinach leaves. We evaluated the synergistic impact of nanobubbles and ultrasound on these two pathogens inactivation in the cell suspension. The results indicated that nanobubbles or ultrasound alone could not significantly reduce bacteria in cell suspension after 15 min. However, a combination of nanobubbles and ultrasonication caused more than 6 log cfu/mL reduction after 15 min, and 7 log cfu/mL reduction after 10 min of *L. innocua* and *E. coli*, respectively. Nanobubbles also enhanced bacterial removal from spinach surface in combination with ultrasonication. Nanobubbles with ultrasound removed more than 2 and 4 log cfu/cm^2^ of *L. innocua* and *E. coli*, respectively, while ultrasound alone caused 0.5 and 1 log cfu/cm^2^ of *L. innocua* and *E. coli* reduction, respectively. No reduction was observed in the solutions with PBS and nanobubbles. Adding food-grade surfactants (0.1% Sodium dodecyl sulfate-SDS, and 0.1% Tween 20), did not significantly enhance nanobubbles efficacy on bacterial removal from spinach surface.

## 1. Introduction

Fresh produce and plant-based materials consumption has been increasing rapidly in the US due to increased consumers’ knowledge about the benefits of fresh produce [1,2]. Fresh produce is one of the reasons for foodborne illness outbreaks in the US [3]. Thus, fresh produce sanitation during post-harvest processing has a critical role in reducing foodborne illnesses [4]. Fresh produce surface properties allow bacteria to be attached to them, increasing food safety risks [5]. Most of these sanitizers are effective in bacterial inactivation in wash water (cell suspension), but their efficacy in reducing bacteria is limited on the fresh produce surfaces. In fresh produce, the unique topography of the leaf surface and the organic content may limit access to sanitizers and wash water [6]. Lack of proper sanitation of fresh produce surfaces can result in cross-contamination risks, increasing the risk of food spoilage and foodborne illnesses. These risks are highly significant for the fresh produce industry since there is no sufficient inactivation step to remove microbes on fresh produce surfaces [7,8]. The efficacy of traditionally used sanitizers such as chlorine is limited due to the presence of organic loads and complexity of the food surfaces, limiting the bacterial reduction to only 1–2 log bacterial on the surface of food [6,9,10,11]. Thus, the detachment of bacteria from the surface of fresh produce will enhance the efficacy of chemical sanitizers and will reduce the required chemicals for inactivating the same number of bacteria when they are on the surface of the fresh produce.

Bacterial detachment from surfaces may depend on different parameters, including mechanical force during washing steps, surface energy, surface tension, and the interaction between bacteria and the plant surface. We recently showed that nanobubble technology has potential for enhancing food safety [4]. Nanobubbles are very stable in solutions due to their nano-size, negative surface charge, and Brownian motion [12,13]. Studies have shown that nanobubbles can reduce contact angle and detach microbial biofilms [4], remove organic materials [14,15,16,17], and dental bacteria [18], detach bacteria from fresh produce [19], and inactivate aquatic pathogens [20]. In our previous studies, we illustrated that ultrasound [20] and chemical sanitizers [4] might enhance the antimicrobial properties of nanobubbles. However, the synergistic properties of nanobubbles, ultrasound, and surfactants have not been evaluated yet. Thus, this study evaluated a novel approach to remove foodborne pathogens from the surface of spinach leaves and inactivate them in the wash water.

## 2. Materials and Methods

### 2.1. Microbial Strains Preparation

We received Listeria innocua (VTE-P1-0002) and Shiga toxin negative *E. coli* O157:H7, from Dr. Laura Strawn (Department of Food Science and Technology, Virginia Tech, Blacksburg, VA, USA) and Dr. Trevor Suslow (Department of Food Science and Technology, the University of California, Davis, CA, USA), respectively. Genetically modified *E. coli* containing a Rifampicin (RIF) resistant plasmid was cultured on a tryptic soy agar (Sigma-Aldrich, St. Louis, MO, USA) with 50 µg/mL RIF. *L. innocua* was cultured on Polymyxin Acriflavine Lithium-chloride Ceftazidime Esculin Mannitol (PALCAM) agar (Merck, Darmstadt, Germany). A single colony of each bacteria was transferred into 10 mL of tryptic soy broth and incubated overnight at 37 °C. After centrifugation of one mL of the broth, the pellet was resuspended in 1 mL sterile PBS to obtain the inoculum with approximately 10^9^ cfu/mL for spinach leaf studies and 10^7^ cfu/mL for cell suspension studies.

### 2.2. Nanobubble Inactivation of Bacteria in Cell Suspension

The nanobubble solution was generated by injecting pure oxygen gas into the water using a nanobubble generator (Moleaer 25 L nanobubble generator, Moleaer Inc., Torrance, CA, USA). We selected pure oxygen based on our preliminary results indicating the effectiveness of pure oxygen in removing microbial biofilms.

The efficacy of nanobubbles alone and in combination with ultrasound against *L. innocua* and *E. coli* was evaluated according to our previous study [20]. Briefly, 9 mL of testing solutions were mixed with one mL of bacteria to obtain a cell suspension with the initial number of 6–7 log cfu/mL and were tested at different times (5 to 15 min) at room temperature. Ultrasound and nanobubbles alone were used separately as the control groups. Bacteria in nanobubble and PBS solutions were placed into the ultrasound bath for 5 to 15 min. Samples were removed and *E. coli* was cultured on (50 µg/mL) RIF-supplemented TSA at 37 °C for 24 h, and *L. innocua* was cultured on PALCAM agar at 37 °C for 48 h.

### 2.3. Washing of Spinach Leaves

In the first experiment, the spinach leaves surface (3 cm^2^) was inoculated by adding 100 µL of inoculums and keeping for 30 min under the hood. Each leaf was immersed into 50 mL of the testing solutions, including nanobubbles and nanobubbles + ultrasound treatments, for 20 min. Samples were recovered after 1, 5, 10, and 20 min. All the experiments were conducted in duplicates and were repeated four times (*n* = 8).

In the second experiment with spinach leaves, we selected two commonly used food-grade surfactants, including Tween-20 and SDS, for this study (Sigma-Aldrich, St. Louis, MO, USA). Different testing solutions with surfactants were prepared by adding Tween-20 and SDS, to the sterile water or nanobubbles solutions to obtain a solution with a final surfactant concentration of 0.1% *w*/*w*. Each leaf was inoculated as mentioned above and was treated with 50 mL of testing solutions for 10 min.

### 2.4. Bacterial Recovery from Spinach Leaves

To recover the bacteria from leaves, 10 mL of maximum recovery diluent (MRD; Oxoid, Basingstoke, UK) was mixed with each leaf and vortexed for 1 min at room temperature. *E. coli* was cultured on (50 µg/mL) RIF-supplemented TSA (37 °C for 24 h), and *L. innocua* was cultured on PALCAM agar (37 °C for 48 h). We also confirmed *L. innocua* inactivation using buffered listeria enrichment broth.

### 2.5. Statistical Analysis

One-way analysis of variance using JMP^®^ Pro 15.0.0 (SAS Institute Inc., Cary, NC, USA) was applied to compare the data at the 5% probability level. All the experiments were conducted four times in duplicate (*n* = 8) (mean ± standard deviation).

## 3. Results and Discussion

### 3.1. Bacterial Inactivation in Cell Suspension

The effect of nanobubbles, ultrasound, and nanobubbles + ultrasound on *L. innocua* and *E. coli* in cell suspension is presented in Figure 1. The results illustrated that both ultrasound and nanobubbles technologies alone did not significantly change bacteria in cell suspension (*p* > 0.05). While, nanobubbles + ultrasound caused more than 3 log cfu/mL reduction in *L. innocua* and *E. coli*, after 5 min. The results showed that *E. coli* was more sensitive to the treatment compared to *L. innocua*. After 10 min, *E. coli* was below the limit of detection, while *L. innocua* was completely inactivated after 15 min.

Nanobubble technology applications in food and agriculture are growing rapidly. However, only a few studies are available on optimizing nanobubbles’ application in inactivating pathogenic bacteria [4,18,19,20]. In our previous studies, we showed that nanobubbles could be used for removing microbial biofilms and inactivating bacteria, including *Vibrio parahaemolyticus*, and *Aeromonas hydrophila* [4,20].

In addition, ultrasound has been used in combination with chemical sanitizers [21,22] for inactivating pathogenic bacteria. In our previous study, we successfully have applied nanobubble technology in combination with ultrasound to inactivate aquaculture pathogens in aquaponics water [20].

Antimicrobial properties of nanobubbles are highly dependent on (1) the physical attributes of the nanobubbles such as gas transfer properties; (2) free hydroxyl radicals generation (OH^•^); (3) releasing a large amount of energy when bubbles burst; and (4) reactive oxygen species (ROS) generation [4,15,23]. When nanobubbles burst by ultrasound on the surface of bacteria, due to their high internal pressure, they release a high amount of energy on the surface, converting oxygen molecules into ROS and causing surface cavitation, resulting in bacterial inactivation [24]. Additionally, collapsing nanobubbles will generate hydroxyl radicals and shock waves in the water [23], which can inactivate bacteria [24]. Our previous study demonstrated the impact of nanobubbles on bacterial DNA alteration, protein oxidation, and cell membrane disruption using vibrational spectroscopy [4].

### 3.2. The Effect of Nanobubbles and Ultrasound on Removing Bacteria from Spinach Leaves

The effect of nanobubbles and ultrasound alone and in combination on removing bacteria from spinach leaves was evaluated by exposing single leaves contaminated with *E. coli* and *L. innocua* to 40 Hz ultrasound for 1, 5, 10, and 20 min at room temperature (Figure 2). The bacterial reduction from the surface of spinach was increased by an increase in ultrasonication time regardless of nanobubbles’ presence and bacterial strain. Similar results were observed in previous studies on removing *E. coli* and *L. innocua* from lettuce surfaces [9], suggesting that differences in bacterial strains based on Gram staining did not significantly impact bacterial removal under ultrasound treatment [9]. For both bacteria, nanobubbles + ultrasonication increased the bacterial removal compared to the ultrasound alone. Nanobubbles + ultrasound caused significantly higher *E. coli* reduction from spinach leaves compared to ultrasound alone, while *L. innocua* reduction by nanobubbles + ultrasound was insignificant in comparison with ultrasound alone. Similar results were observed in our previous studies on removing *E. coli* and *L. innocua* mono-species biofilms from stainless steel and plastic coupons [4]. In our previous study, we found that *L. innocua* biofilm was more resistant to nanobubbles than *E. coli.*

### 3.3. Effect of Surfactants on the Efficacy of Nanobubbles

To determine the impact of different surfactants (0.1% SDS and 0.1% Tween 20) on the efficacy of nanobubbles alone and in combination with ultrasound on the removal of bacteria from spinach surface, inoculated spinach leaves were exposed to different treatments (Figure 3). Solutions with 0.1% of each surfactant in DI water resulted in some bacterial detachment (0.3–0.7 log cfu/cm^2^). Surfactants did not improve bacterial reduction when combined with nanobubbles, or ultrasound compared to the treatments without surfactants. Combining all the treatments, including nanobubbles + ultrasound + surfactants also did not improve the bacterial reduction compared to nanobubbles + ultrasound. The results indicate that surfactants do not have a significant impact on the efficacy of nanobubbles and ultrasound alone or in combination on the bacterial removal from spinach surfaces. The improvement in removing bacteria from different surfaces using surfactants has been shown by several researchers [9]. Technically, the bacterial removal enhancement by surfactants is related to the decrease in the contact angle of the fresh produce surface [9]. In addition, nanobubbles can reduce the surface contact angle [4,9]. It has been shown that adding surfactants into the nanobubble solutions does not have a negative impact on the physical appearance and stability of the nanobubbles [25]. Thus, our results from this study indicate that nanobubbles are stable in the presence of surfactants, and their bacterial removal efficacy does not depend on surfactants, resulting in novel chemical-free approaches for removing bacteria from surfaces.

## 4. Conclusions

In this study, we determined the antimicrobial properties of nanobubbles and ultrasound against *L. innocua* and *E. coli* in cell suspension, as well as the effect of different surfactants and nanobubbles on the removal of pathogenic bacteria from the spinach leaves. Nanobubbles, or ultrasound alone did not reduce the bacteria in cell suspension, while a combination of nanobubbles and ultrasound resulted in complete reduction in *E. coli*, and *L. innocua* after 10 and 15 min, respectively. Nanobubbles and ultrasound did not reduce bacteria on the spinach leaf surface significantly, while combining them together resulted in 2, and 4 log cfu/cm^2^ reductions in *L. innocua* and *E. coli*, respectively. Adding two food-grade surfactants, including 0.1% SDS and 0.1% Tween 20 did not enhance the removal efficacy of nanobubbles, nanobubbles + ultrasound, or ultrasound. More experiments are required to determine the impact of surfactants on the properties and stability of nanobubbles.

## Figures and Tables

**Figure 1 foods-10-02154-f001:**
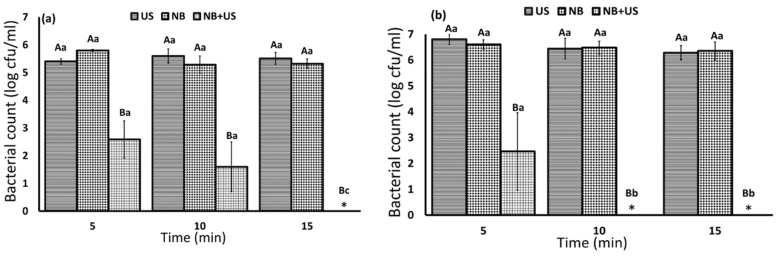
(**a**) *L. innocua* and (**b**) *E. coli* counts in nanobubbles (NB), ultrasound (US) and nanobubbles + ultrasound (NB+US) treated samples at different exposure times. Statistical difference was determined based on *p* < 0.05 (*). Limit of detection was below 0.5 log cfu/mL. The initial number of bacteria was 6.2 and 7.18 cfu/mL for *L. innocua* and *E. coli*, respectively. Capital letters indicate the significant differences among the treatments at given time, and lower case letters indicate the significant differences of specific treatment at different times.

**Figure 2 foods-10-02154-f002:**
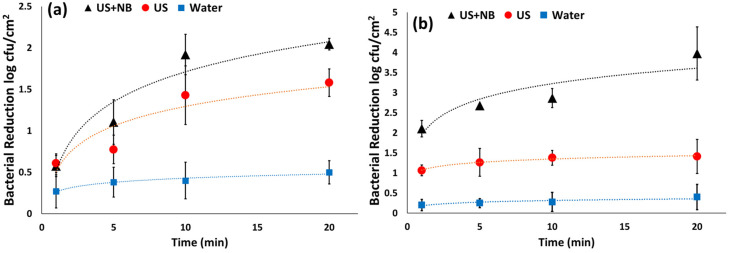
(**a**) *L. innocua* and (**b**) *E. coli* reduction in ultrasound (US) and nanobubbles + ultrasound (NB + US) treated spinach leaves at different exposure times.

**Figure 3 foods-10-02154-f003:**
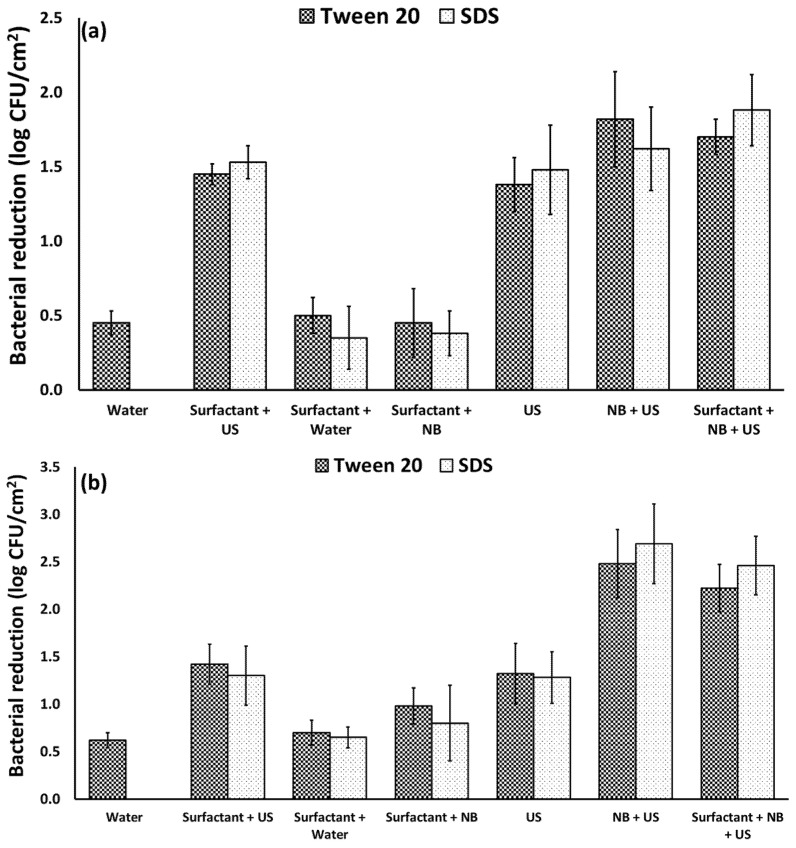
(**a**) *L. innocua* and (**b**) *E. coli* reduction on spinach leaves treated with different solutions for 10 min.

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
