# Peer review of "The Effect Ultrasound and Surfactants on Nanobubbles Efficacy against Listeria innocua and Escherichia coli O157:H7, in Cell Suspension and on Fresh Produce Surfaces"

_foods, 2021, doi:10.3390/foods10092154_

Round 1

Reviewer 1 Report

The study evaluated the effect of ultrasound and surfactants on nanobubbles efficacy against Listeria innocua and Escherichia coli O157:H7, in cell suspension and on fresh produce surfaces. The study is relevant but the methodology is poorly designed. No control or baseline data have been presented and the methodology is not clearly explained. The discussion section also needs a lot of work to bring it to good shape.

L-5: produce surfaces (add s)

L 18-19: Please mention the time that took to result in a reduction by 6 long and 7 log.

L 72-76: Needs more elaboration on how the nanobubbles and ultrasounds were used? What was the model and brand name of that equipment? The treatment method is not clear.

L-79-87: The procedure is not clear. Did you dip spinach in the solution or just poured the solution on the leaves?

L-107-108: How did you make sure that there was a complete inactivation of L. innocua? Did you test it by enrichment in enrichment broth followed by plating?

L110 Fig 1: There is no data of 0 minutes shown in the figures. It is hard to compare the reduction if you do not have the baseline data. In addition, there is no control, i.e. the inoculated samples without any treatment.

L112: fig 1: Please add the type of the sample on the caption.  

Figure 2: Y-axis is supposed to be log CFU . The same concern here, no baseline data was presented. The authors need to show 0 min data before showing the reduction over time. They have presented data for 1, 5, 10 20 mins only. The results have been presented per cm2 spinach. In the methodology section, it is not clear what area of spinach was inoculated and what was the size of the spinach samples.

No data were presented on the effect of nanobubbles only for spinach samples. The reduction in the population may be due to washing off the leaves in the solution.  To determine it, the authors should have analyzed the spent solution.

Figure 3: write down the treatment time., 20 mins?

Author Response

Thank you for providing great comments.

L-5: produce surfaces (add s)

Response: Revised.

L 18-19: Please mention the time that took to result in a reduction by 6 long and 7 log.

Response: Revised.

L 72-76: Needs more elaboration on how the nanobubbles and ultrasounds were used? What was the model and brand name of that equipment? The treatment method is not clear.

Response: Revised.

L-79-87: The procedure is not clear. Did you dip spinach in the solution or just poured the solution on the leaves?

Response: Revised. Correct, we dip the spinach leaf into the solution.  

L-107-108: How did you make sure that there was a complete inactivation of L. innocua? Did you test it by enrichment in enrichment broth followed by plating?

Response: We waited for 48 h and also enriched the samples. Did not observe any growth.

L110 Fig 1: There is no data of 0 minutes shown in the figures. It is hard to compare the reduction if you do not have the baseline data. In addition, there is no control, i.e. the inoculated samples without any treatment.

Response: The initial number of the bacteria was added to the figure caption. During the 15 min, the bacteria number in a PBS solution did not change. We did not bring the water/PBS to keep the figure less busy.

L112: fig 1: Please add the type of the sample on the caption.

Response: Revised.

Figure 2: Y-axis is supposed to be log CFU . The same concern here, no baseline data was presented. The authors need to show 0 min data before showing the reduction over time. They have presented data for 1, 5, 10 20 mins only. The results have been presented per cm2 spinach. In the methodology section, it is not clear what area of spinach was inoculated and what was the size of the spinach samples.

Response: Revised. This figure is about bacterial reduction and at 0 time, there is no bacterial reduction and it is technically 0. In addition, the baseline here is bacteria on spinach in just water solution, which technically no reduction was observed.

No data were presented on the effect of nanobubbles only for spinach samples. The reduction in the population may be due to washing off the leaves in the solution.  To determine it, the authors should have analyzed the spent solution.

Response: Revised. We have measured the number of bacteria in spent solution as well. Since the solution was exposed to the US as well, no bacteria was observed. Technically, the goal here is removing bacteria from the surface and bring them into the solution.

Figure 3: write down the treatment time., 20 mins?

Response: Revised.

Reviewer 2 Report

The topic is of interest to this journal's readership and the novel technology is an interesting approach to produce safety. Please see minor comments below. Additionally, there are some minor grammatical flaws throughout that should be addressed. 

Abstract - please state quantitative results

L17: Could not be reduced at all or not significantly reduced? 

L20: Please quantify the reduction achieved. 

Introduction - no comments

Methods

L74: The authors previously indicated that the PBS suspensions were ~10^9 CFU/ml. Here they state that after a 1:10 dilution, counts are 10^6-10^7, what accounts for the difference? 

L81: Meaning, two leaves per trial and a total of four trials? Were the duplicates within a trial treated as equal replicates to data from other trials and do these represent equal levels of variation?

Results

L150: How do these levels of reduction compare to washing with water alone?

L156: Are these surfactants food grade?

Author Response

Abstract - please state quantitative results

Response: Revised.

L17: Could not be reduced at all or not significantly reduced? 

Response: Revised.

L20: Please quantify the reduction achieved. 

Response: Revised.

Introduction - no comments

Methods

L74: The authors previously indicated that the PBS suspensions were ~10^9 CFU/ml. Here they state that after a 1:10 dilution, counts are 10^6-10^7, what accounts for the difference? 

Response: Revised. The 10^9 was used for inoculating the spinach leaves, and 10^7 was used for cell suspension study.

L81: Meaning, two leaves per trial and a total of four trials? Were the duplicates within a trial treated as equal replicates to data from other trials and do these represent equal levels of variation?

Response: Each time, we had two samples/spinach leaves for each experiment (duplicates). Then we repeated the whole experiment four times. Variation from each experiment was close to other experiment. Each experiment started by transferring new colony from TSA to TSB.

Results

L150: How do these levels of reduction compare to washing with water alone?

Response: Bacterial reduction in just water was minimal. We revised the graph and added the results.  

L156: Are these surfactants food grade?

Response: Revised. Yes, they are all food grade.  

Round 2

Reviewer 1 Report

Addressed most of the comments.